# Independence clustering (without a matrix)

**Daniil Ryabko**
INRIA Lillle,
40 avenue de Halley, Villeneuve d'Ascq, France
daniil@ryabko.net

## Abstract

The independence clustering problem is considered in the following formulation: given a set $S$ of random variables, it is required to find the finest partitioning $\{U_1, \ldots, U_k\}$ of $S$ into clusters such that the clusters $U_1, \ldots, U_k$ are mutually independent. Since mutual independence is the target, pairwise similarity measurements are of no use, and thus traditional clustering algorithms are inapplicable. The distribution of the random variables in $S$ is, in general, unknown, but a sample is available. Thus, the problem is cast in terms of time series. Two forms of sampling are considered: i.i.d. and stationary time series, with the main emphasis being on the latter, more general, case. A consistent, computationally tractable algorithm for each of the settings is proposed, and a number of fascinating open directions for further research are outlined.

## 1 Introduction

Many applications face the situation where a set $S = \{\mathbf{x}_1, \ldots, \mathbf{x}_N\}$ of samples has to be divided into clusters in such a way that inside each cluster the samples are dependent, but the clusters between themselves are as independent as possible. Here each $\mathbf{x}_i$ may itself be a sample or a time series $\mathbf{x}_i = X_1^i, \ldots, X_n^i$. For example, in financial applications, $\mathbf{x}_i$ can be a series of recordings of prices of a stock $i$ over time. The goal is to find the segments of the market such that different segments evolve independently, but within each segment the prices are mutually informative [15, 17]. In biological applications, each $\mathbf{x}_i$ may be a DNA sequence, or may represent gene expression data [28, 20], or, in other applications, an fMRI record [4, 13].

The staple approach to this problem in applications is to construct a matrix of (pairwise) correlations between the elements, and use traditional clustering methods, e.g., linkage-based methods or $k$ means and its variants, with this matrix [15, 17, 16]. If mutual information is used, it is used as a (pairwise) proximity measure between individual inputs, e.g. [14].

We remark that pairwise independence is but a surrogate for (mutual) independence, and, in addition, correlation is a surrogate for pairwise independence. There is, however, no need to resort to surrogates unless forced to do so by statistical or computational hardness results. We therefore propose to reformulate the problem from the first principles, and then show that it is indeed solvable both statistically and computationally — but calls for completely different algorithms. The formulation proposed is as follows.

*Given a set $S = (\mathbf{x}_1, \ldots, \mathbf{x}_N)$ of random variables, it is required to find the finest partitioning $\{U_1, \ldots, U_k\}$ of $S$ into clusters such that the clusters $U_1, \ldots, U_k$ are mutually independent.*

To our knowledge, this problem in its full generality has not been addressed before. A similar informal formulation appears in the work [1] that is devoted to optimizing a generalization of the ICA objective. However, the actual problem considered only concerns the case of tree-structured dependence, which allows for a solution based on pairwise measurements of mutual information.

Note that in the fully general case pairwise measurements are useless, as are, furthermore, bottom-up (e.g., linkage-based) approaches. Thus, in particular, a proximity matrix cannot be used for the analysis. Indeed, it is easy to construct examples in which any pair or any small group of elements are independent, but are dependent when the same group is considered jointly with more elements. For instance, consider a group of Bernoulli 1/2-distributed random variables $\mathbf{x}_1, \ldots, \mathbf{x}_{N+1}$, where $\mathbf{x}_1, \ldots, \mathbf{x}_N$ are i.i.d. and $\mathbf{x}_{N+1} = \sum_{i=1}^{N} \mathbf{x}_i \bmod 2$. Note that any $N$ out of these $N + 1$ random variables are i.i.d., but together the $N + 1$ are dependent. Add then two more groups like this, say, $\mathbf{y}_1, \ldots, \mathbf{y}_{N+1}$ and $\mathbf{z}_1, \ldots, \mathbf{z}_{N+1}$ that have the exact same distribution, with the groups of $\mathbf{x}$, $\mathbf{y}$ and $\mathbf{z}$ mutually independent. Naturally, these are the three clusters we would want to recover. However, if we try to cluster the union of the three, then any algorithm based on pairwise correlations will return an essentially arbitrary result. What is more, if we try to find *clusters* that are pairwise independent, then, for example, the clustering $\{(\mathbf{x}_i, \mathbf{y}_i, \mathbf{z}_i)_{i=1..N}\}$ of the input set into $N + 1$ clusters appears correct, but, in fact, the resulting clusters are dependent. Of course, real-world data does not come in the form of summed up Bernoulli variables, but this simple example shows that considering independence of small subsets may be very misleading.

The considered problem is split into two parts considered separately: the computational and the statistical part. This is done by first considering the problem assuming the joint distribution of all the random variables is known, and is accessible via an oracle. Thus, the problem becomes computational. A simple, computationally efficient algorithm is proposed for this case. We then proceed to the time-series formulations: the distribution of $(\mathbf{x}_1, \ldots, \mathbf{x}_N)$ is unknown, but a sample $(X_1^1, \ldots, X_1^N), \ldots, (X_n^1, \ldots, X_n^N)$ is provided, so that $\mathbf{x}_i$ can be identified with the time series $X_1^i, \ldots, X_n^i$. The sample may be either independent and identically distributed (i.i.d.), or, in a more general formulation, stationary. As one might expect, relying on the existing statistical machinery, the case of known distributions can be directly extended to the case of i.i.d. samples. Thus, we show that it is possible to replace the oracle access with statistical tests and estimators, and then use the same algorithm as in the case of known distributions. The general case of stationary samples turns out to be much more difficult, in particular because of a number of strong impossibility results. In fact, it is challenging already to determine what is possible and what is not from the statistical point of view. In this case, it is not possible to replicate the oracle access to the distribution, but only its weak version that we call *fickle oracle*. We find that, in this case, it is only possible to have a consistent algorithm for the case of known $k$. An algorithm that has this property is constructed. This algorithm is computationally feasible when the number of clusters $k$ is small, as its complexity is $O(N^{2k})$. Besides, a measure of information divergence is proposed for time-series distributions that may be of independent interest, since it can be estimated consistently without any assumptions at all on the distributions or their densities (the latter may not exist).

**The main results** of this work are theoretical. The goal is to determine, as a first step, what is possible and what is not from both statistical and computational points of view. The main emphasis is placed on highly dependent time series, as warranted by the applications cited above, leaving experimental investigations for future work. The contribution can be summarized as follows:

- a consistent, computationally feasible algorithm for known distributions, unknown number of clusters, and an extension to the case of **unknown distributions** and i.i.d. samples;
- an algorithm that is consistent under stationary ergodic sampling with arbitrary, unknown distributions, but with a known number $k$ of clusters;
- an impossibility result for clustering stationary ergodic samples with $k$ unknown;
- an information divergence measure for stationary ergodic time-series distributions along with its estimator that is consistent without any extra assumptions;

In addition, an array of open problems and exciting directions for future work is proposed.

**Related work.** Apart from the work on independence clustering mentioned above, it is worth pointing out the relation to some other problems. First, the proposed problem formulation can be viewed as a Bayesian-network learning problem: given an unknown network, it is required to split it into independent clusters. In general, learning a Bayesian network is NP-hard [5], even for rather restricted classes of networks (e.g., [18]). Here the problem we consider is much less general, which is why it admits a polynomial-time solution. A related clustering problem, proposed in [23] (see also [12]) is clustering time series with respect to distribution. Here, it is required to put two time series samples $\mathbf{x}_1, \mathbf{x}_2$ into the same cluster if and only if their distribution is the same. Similar to the independence clustering introduced here, this problem admits a consistent algorithm if the samples are i.i.d. (or

mixing) and the number of distributions (clusters) is unknown, and in the case of stationary ergodic samples if and only if $k$ is known.

## 2  Set-up and preliminaries

A set $S := \{\mathbf{x}_1, \ldots, \mathbf{x}_N\}$ is given, where we will either assume that the joint distribution of $\mathbf{x}_i$ is known, or else that the distribution is unknown but a sample $(X_1^1, \ldots, X_n^1), \ldots, (X_1^N, \ldots, X_n^N)$ is given. In the latter case, we identify each $\mathbf{x}_i$ with the sequence (sample) $X_1^i, \ldots, X_n^i$, or $X_{1..n}^i$ for short, of length $n$. The lengths of the samples are the same only for the sake of notational convenience; it is easy to generalize all algorithms to the case of different sample lengths $n_i$, but the asymptotic would then be with respect to $n := \min_{i=1..N} n_i$. It is assumed that $X_j^i \in \mathcal{X} := \mathbb{R}$ are real-valued, but extensions to more general cases are straightforward.

For random variables $A, B, C$ we write $(A \perp B)|C$ to say that $A$ is conditionally independent of $B$ given $C$, and $A \perp B \perp C$ to say that $A, B$ and $C$ are mutually independent.

The (unique up to a permutation) partitioning $\mathbf{U} := \{U_1, \ldots, U_k\}$ of the set $S$ is called the **ground-truth clustering** if $U_1, \ldots, U_k$ are mutually independent ($U_1 \perp \cdots \perp U_k$) and no refinement of $\mathbf{U}$ has this property. A **clustering algorithm is consistent** if it outputs the ground-truth clustering, and it is asymptotically consistent if w.p. 1 it outputs the ground-truth clustering from some $n$ on.

For a discrete $A$-valued r.v. $X$ its Shannon entropy is defined as $H(X) := \sum_{a \in A} -P(X = a) \log P(X = a)$, letting $0 \log 0 = 0$. For a distribution with a density $f$ its *(differential) entropy* is defined as $H(X) =: -\int f(x) \log f(x)$. For two random variables $X, Y$ their *mutual information* $I(X, Y)$ is defined as $I(X, Y) = H(X) + H(Y) - H(X, Y)$. For discrete random variables, as well as for continuous ones with a density, $X \perp Y$ if and only if $I(X, Y) = 0$; see, e.g., [6]. Likewise, $I(X_1, \ldots, X_m)$ is defined as $\sum_{i=1..m} H(X_i) - H(X_1, \ldots, X_m)$.

For the sake of convenience, in the next two sections we make the assumption stated below. However, we will show (Sections 5,6) that this assumption can be gotten rid of as well.

**Assumption 1.** All distributions in question have densities bounded away from zero on their support.

## 3  Known distributions

As with any statistical problem, it is instructive to start with the case where the (joint) distribution of all the random variables in question is known. Finding out what can be done and how to do it in this case helps us to set the goals for the (more realistic) case of unknown distributions.

Thus, in this section, $\mathbf{x}_1, \ldots, \mathbf{x}_N$ are not time series, but random variables whose joint distribution is known to the statistician. The access to this distribution is via an oracle; specifically, our oracle will provide answers to the following questions about mutual information (where, for convenience, we assume that the mutual information with the empty set is 0):
**Oracle TEST.** Given sets of random variables $A, B, C, D \subset \{\mathbf{x}_1, \ldots, \mathbf{x}_N\}$ answer whether $I(A, B) > I(C, D)$.

**Remark 1** ( Conditional independence oracle). Equivalently, one can consider an oracle that answers *conditional independence* queries of the form $(A \perp B)|C$. The definition above is chosen for the sake of continuity with the next section, and it also makes the algorithm below more intuitive. However, in order to test conditional independence statistically one does not have to use mutual information, but may resort to any other divergence measure instead.

The proposed algorithm (see the pseudocode listing below) works as follows. It attempts to split the input set recursively into two independent clusters, until it is no longer possible. To split a set in two, it starts with putting one element $\mathbf{x}$ from the input set $S$ into a candidate cluster $C := \{\mathbf{x}\}$, and measures its mutual information $I(C, R)$ with the rest of the set, $R := S \setminus C$. If $I(C, R)$ is already 0 then we have split the set into two independent clusters and can stop. Otherwise, the algorithm then takes the elements out of $R$ one by one *without replacement* and each time looks whether $I(C, R)$ has decreased. Once such an element is found, it is moved from $R$ to $C$ and *the process is restarted* from the beginning with $C$ thus updated. Note that, if we have started with $I(C, R) > 0$, then taking elements out of $R$ without replacement we eventually should find a one that decreases $I(C, R)$, since $I(C, \varnothing) = 0$ and $I(C, R)$ cannot increase in the process.

**Theorem 1.** *The algorithm CLIN outputs the correct clustering using at most $2kN^2$ oracle calls.*

*Proof.* We shall first show that the procedure for splitting a set into two indeed splits the input set into two independent sets, if and only if such two sets exist. First, note that if $I(C, S \setminus C) = 0$ then $C \perp R$ and the function terminates. In the opposite case, when $I(C, S \setminus C) > 0$, by removing an element from $R := S \setminus C$, $I(C, R)$ can only decrease (indeed, $h(C|R) \leq h(C|R \setminus \{x\})$ by information processing inequality). Eventually when all elements are removed, $I(C, R) = I(C, \{\}) = 0$, so there must be an element $\mathbf{x}$ removing which decreases $I(C, R)$. When such an element $\mathbf{x}$ found it is moved to $C$. Note that, in this case, indeed $\mathbf{x} \not\perp C$. However, it is possible that removing an element $\mathbf{x}$ from $R$ does not reduce $I(C, R)$, yet $\mathbf{x} \not\perp C$. This is why the `while` loop is needed, that is, the whole process has to be repeated until no elements can be moved to $C$. By the end of each `for` loop, we have either found at least one element to move to $C$, or we have assured that $C \perp S \setminus C$ and the loop terminates. Since there are only finitely many elements in $S \setminus C$, the `while` loop eventually terminates. Moreover, each of the two loops (`while` and `for`) terminates in at most $N$ iterations. Finally, notice that if $(C_1, C_2) \perp C_3$ and $C_1 \perp C_2$ then also $C_1 \perp C_2 \perp C_3$, which means that by repeating the Split function recursively we find the correct clustering.

From the above, the bound on the number of oracle calls is easily obtained by direct calculation.    □

## 4 I.I.D. sampling

Figure 1: CLIN: cluster with $k$ unknown, given an oracle for MI

    INPUT: The set $S$.
    $(C_1, C_2) := Split(S)$
    **if** $C_2 \neq \varnothing$ **then**
        Output:$CLIN(C_1), CLIN(C_2)$
    **else**
        Output: $C_1$
    **end if**
    **Function Split**(Set $S$ of samples)
    Initialize: $C := \{\mathbf{x}_1\}$, $R := S \setminus C$;
    **while** TEST$(I(C; R) > 0)$ **do**
        **for** each $\mathbf{x} \in R$ **do**
            **if** TEST$(I(C; R) > I(C; R \setminus \{\mathbf{x}\}))$
            **then**
                move $\mathbf{x}$ from $R$ to $C$
                break the **for** loop
            **else**
                move $\mathbf{x}$ from $R$ to $M$
            **end if**
        **end for**
        $M := \{\}$, $R := S \setminus C$;
    **end while**
    Return$(C, R)$
    **END function**

In this section we assume that the distribution of $(\mathbf{x}_1, \ldots, \mathbf{x}_N)$ is not known, but an i.i.d. sample $(X_1^1, \ldots, X_1^N), \ldots, (X_n^1, \ldots, X_n^N)$ is provided. We identify $\mathbf{x}_i$ with the (i.i.d.) time series $X_{1..n}^i$. Formally, $N$ $\mathcal{X}$-valued processes is just a single $\mathcal{X}^N$-valued process. The latter can be seen as a matrix $(X_j^i)_{i=1..N, j=1..\infty}$, where each row $i$ is the sample $\mathbf{x}_i = X_{1..n}^i$ and each column $j$ is what is observed at time $j$: $X_j^1 .. X_j^N$.

The case of i.i.d. samples is not much different from the case of a known distribution. What we need is to replace the oracle test with (nonparametric) statistical tests. First, a test for independence is needed to replace the oracle call TEST$(I(C, R) > 0)$ in the `while` loop. Such tests are indeed available, see, for example, [8]. Second, we need an estimator of mutual information $I(X, Y)$, or, which is sufficient, for entropies, but with a rate of convergence. If the rate of convergence is known to be asymptotically bounded by, say, $t(n)$, then, in order to construct an asymptotically consistent test, we can take any $t'(n) \to 0$ such that $t(n) = o(t'(n))$ and decide our inequality as follows: if $\hat{I}(C; R \setminus \{\mathbf{x}\}) < \hat{I}(C; R) - t'(n)$ then say that $I(C; R \setminus \{\mathbf{x}\}) < I(C; R)$. The required rates of convergence, which are of order $\sqrt{n}$ under Assumption 1, can be found in [3].

Given the result of the previous section, it is clear that if the oracle is replaced by the tests described, then CLIN is a.s. consistent. Thus, we have demonstrated the following.

**Theorem 2.** *Under Assumption 1, there is an asymptotically consistent algorithm for independence clustering with i.i.d. sampling.*

**Remark 2** (**Necessity of the assumption**). The independence test of [8] does not need Assumption 1, as it is distribution-free. Since the mutual information is defined in terms of densities, if we want to completely get rid of Assumption 1, we would need to use some other measure of dependence for the test. One such measure is defined in the next section already for the general case of process distributions. However, the rates of convergence of its empirical estimates under i.i.d. sampling remain to be studied.

**Remark 3** (**Estimators vs. tests**). As noted in Remark 1, the tests we are using are, in fact, tests for (conditional) independence: testing $I(C; R) > I(C; R \setminus \{\mathbf{x}\})$ is testing for $(C \perp \{\mathbf{x}\}|R \setminus \{\mathbf{x}\})$. Conditional independence can be tested directly, without estimating $I$ (see, for example 27), potentially allowing for tighter performance guarantees under more general conditions.

## 5 Stationary sampling

We now enter the hard mode. The general case of stationary sampling presents numerous obstacles, often in the form of theoretical impossibility results: there are (provably) no rates of convergence, no independence test, and 0 mutual information rate does not guarantee independence. Besides, some simple-looking questions regarding the existence of consistent tests, which indeed have simple answers in the i.i.d. case, remain open in the stationary ergodic case. Despite all this, a computationally feasible asymptotically consistent independence clustering algorithm can be obtained, although only for the case of a known number of clusters. This parallels the situation of clustering according to the distribution [23, 12].

In this section we assume that the distribution of $(\mathbf{x}_1, \ldots, \mathbf{x}_N)$ is not known, but a jointly stationary ergodic sample $(X_1^1, \ldots, X_1^N), \ldots, (X_n^1, \ldots, X_n^N)$ is provided. Thus, $\mathbf{x}_i$ is a stationary ergodic time series $X_{1..n}^i$. Here is also where we **drop Assumption 1**; in particular, densities do not have to exist. This new relaxed set of assumptions can be interpreted as using a weaker oracle, as explained in Remark 5 below.

We start with preliminaries about stationary processes, followed by impossibility results, and concluding with an algorithm for the case of known $k$.

### 5.1 Preliminaries: stationary ergodic processes

**Stationary, ergodicity, information rate.** (Time-series) distributions, or processes, are measures on the space $(\mathcal{X}^\infty, \mathcal{F}_{\mathcal{X}^\infty})$, where $\mathcal{F}_{\mathcal{X}^\infty}$ is the Borel sigma-algebra of $\mathcal{X}^\infty$. Recall that $N$ $\mathcal{X}$-valued process is just a single $\mathcal{X}^N$-valued process. So the distributions are on the space $((\mathcal{X}^N)^\infty, \mathcal{F}_{(A^N)^\infty})$; this will be often left implicit. For a sequence $\mathbf{x} \in A^n$ and a set $B \in \mathcal{B}$ denote $\nu(\mathbf{x}, B)$ the *frequency* with which the sequence $\mathbf{x}$ falls in the set $B$. A process $\rho$ is *stationary* if $\rho(X_{1..|B|} = B) = \rho(X_{t..t+|B|-1} = B)$ for any measurable $B \in \mathcal{X}^*$ and $t \in \mathbb{N}$. We further abbreviate $\rho(B) := \rho(X_{1..|B|} = B)$. A stationary process $\rho$ is called *(stationary) ergodic* if the frequency of occurrence of each measurable $B \in \mathcal{X}^*$ in a sequence $X_1, X_2, \ldots$ generated by $\rho$ tends to its a priori (or limiting) probability a.s.: $\rho(\lim_{n \to \infty} \nu(X_{1..n}, B) = \rho(B)) = 1$. By virtue of the ergodic theorem, this definition can be shown to be equivalent to the more standard definition of stationary ergodic processes given in terms of shift-invariant sets [26]. Denote $\mathcal{S}$ and $\mathcal{E}$ the sets of all stationary and stationary ergodic processes correspondingly. The **ergodic decomposition** theorem for stationary processes (see, e.g., 7) states that any stationary process can be expressed as a mixture of stationary ergodic processes. That is, a stationary process $\rho$ can be envisaged as first selecting a stationary ergodic distribution according to some measure $W_\rho$ over the set of all such distributions, and then using this ergodic distribution to generate the sequence. More formally, for any $\rho \in \mathcal{S}$ there is a measure $W_\rho$ on $(\mathcal{S}, \mathcal{F}_{\mathcal{S}})$, such that $W_\rho(\mathcal{E}) = 1$, and $\rho(B) = \int dW_\rho(\mu)\mu(B)$, for any $B \in \mathcal{F}_{\mathcal{X}^\infty}$.

For a stationary time series $\mathbf{x}$, its *$m$-order entropy* $h_m(\mathbf{x})$ is defined as $\mathbf{E}_{X_{1..m-1}}h(X_m|X_{1..m-1})$ (so the usual Shannon entropy is the entropy of order 0). By stationarity, the limit $\lim_{m \to \infty} h_m$ exists and equals $\lim_{m \to \infty} \frac{1}{m}h(X_{1..m})$ (see, for example, [6] for more details). This limit is called *entropy rate* and is denoted $h_\infty$. For $l$ stationary processes $\mathbf{x}_i = (X_1^i, \ldots, X_n^i, \ldots)$, $i = 1..l$, the $m$-order mutual information is defined as $I_m(\mathbf{x}_1, \ldots, \mathbf{x}_l) := \sum_{i=1}^l h_m(x_i) - h_m(\mathbf{x}_1, \ldots, \mathbf{x}_l)$ and the *mutual information rate* is defined as the limit

$$I_\infty(\mathbf{x}_1, \ldots, \mathbf{x}_l) := \lim_{m \to \infty} I_m(\mathbf{x}_1, \ldots, \mathbf{x}_l). \tag{1}$$

**Discretisations and a metric.** For each $m, l \in \mathbb{N}$, let $B^{m,l}$ be a partitioning of $\mathcal{X}^m$ into $2^l$ sets such that jointly they generate $\mathcal{F}_m$ of $\mathcal{X}^m$, i.e. $\sigma(\cup_{l \in \mathbb{N}} B^{m,l}) = \mathcal{F}_m$. The distributional distance between a pair of process distributions $\rho_1, \rho_2$ is defined as follows [7]:

$$d(\rho_1, \rho_2) = \sum_{m,l=1}^\infty w_m w_l \sum_{B \in B^{m,l}} |\rho_1(B) - \rho_2(B)|, \tag{2}$$

where we set $w_j := 1/j(j+1)$, but any summable sequence of positive weights may be used. As shown in [22], empirical estimates of this distance are asymptotically consistent for arbitrary stationary ergodic processes. These estimates are used in [23, 12] to construct time-series clustering algorithms for clustering with respect to distribution. Here we will only use this distance in the impossibility results. Basing on these ideas, Györfi [9] suggested to use a similar construction for studying independence, namely $d(\rho_1, \rho_2) = \sum_{m,l=1}^{\infty} w_m w_l \sum_{A,B \in B^{m,l}} |\rho_1(A)\rho_2(B) - \rho(A \times B)|$, where $\rho_1$ and $\rho_2$ are the two marginals of a process $\rho$ on pairs, and noted that its empirical estimates are asymptotically consistent. The distance we will use is similar, but is based on mutual information.

## 5.2 Impossibility results

First of all, while the definition of ergodic processes guarantees convergence of frequencies to the corresponding probabilities, this convergence can be arbitrary slow [26]: there are no meaningful bounds on $|\nu(X_{1..n}, 0) - \rho(X_1 = 0)|$ in terms of $n$ for ergodic $\rho$. This means that we cannot use tests for (conditional) independence of the kind employed in the i.i.d. case (Section 4).

Thus, the first question we have to pose is whether it is possible to test independence, that is, to say whether $\mathbf{x}_1 \perp \mathbf{x}_2$ based on a stationary ergodic samples $X_{1..n}^1, X_{1..n}^2$. Here we show that the answer in a certain sense is negative, but some important questions remain open.

An (independence) test $\varphi$ is a function that takes two samples $X_{1..n}^1, X_{1..n}^2$ and a parameter $\alpha \in (0,1)$, called the *confidence level*, and outputs a binary answer: independent or not. A test $\varphi$ is $\alpha$-*level consistent* if, for every stationary ergodic distribution $\rho$ over a pair of samples $(X_{1..n..}^1, X_{1..n..}^2)$, for every confidence level $\alpha$, $\rho(\varphi_\alpha(X_{1..n}^1, X_{1..n}^2) = 1) < \alpha$ if the marginal distributions of the samples are independent, and $\varphi_\alpha(X_{1..n}^1, X_{1..n}^2)$ converges to 1 as $n \to \infty$ with $\rho$-probability 1 otherwise.

The next proposition can be established using the criterion of [25]. Recall that, for $\rho \in \mathcal{S}$, the measure $W_\rho$ over $\mathcal{E}$ is its ergodic decomposition. The criterion states that there is an $\alpha$-level consistent test for $H_0$ against $\mathcal{E} \setminus H_0$ if an only if $W_\rho(H_0) = 1$ for every $\rho \in \mathrm{cl}\, H_0$.

**Proposition 1.** *There is no $\alpha$-level consistent independence test (jointly stationary ergodic samples).*

*Proof.* The example is based on the so-called translation process, constructed as follows. Fix some irrational $\alpha \in (0,1)$ and select $r_0 \in [0,1]$ uniformly at random. For each $i = 1..n..$ let $r_i = (r_{i-1} + \alpha) \bmod 1$ (the previous element is shifted by $\alpha$ to the right, considering the [0,1] interval looped). The samples $X_i$ are obtained from $r_i$ by thresholding at $1/2$, i.e. $X_i := \mathbb{I}\{r_i > 0.5\}$ (here $r_i$ can be considered hidden states). This process is stationary and ergodic; besides, it has 0 entropy rate [26], and this is not the last of its peculiarities. Take now two independent copies of this process to obtain a pair $(\mathbf{x}_1, \mathbf{x}_2) = (X_1^1, X_1^2 \ldots, X_n^1, X_n^2, \ldots)$. The resulting process on pairs, which we denote $\rho$, is stationary, but it is not ergodic. To see the latter, observe that the difference between the corresponding hidden states remains constant. In fact, each initial state $(r_1, r_2)$ corresponds to an ergodic component of our process on pairs. By the same argument, these ergodic components are not independent. Thus, we have taken two independent copies of a stationary ergodic process, and obtained a stationary process which is not ergodic and whose ergodic components are pairs of processes that are not independent! To apply the criterion cited above, it remains to show that the process $\rho$ we constructed can be obtained as a limit of stationary ergodic processes on pairs. To see this, consider, for each $\varepsilon$, a process $\rho_\varepsilon$, whose construction is identical to $\rho$ except that instead of shifting the hidden states by $\alpha$ we shift them by $\alpha + u_i^\varepsilon$ where $u_i^\varepsilon$ are i.i.d. uniformly random on $[-\varepsilon, \varepsilon]$. It is easy to see that $\lim_{\varepsilon \to 0} \rho_\varepsilon = \rho$ in distributional distance, and all $\rho_\varepsilon$ are stationary ergodic. Thus, if $H_0$ is the set of all stationary ergodic distributions on pairs, we have found a distribution $\rho \in \mathrm{cl}\, H_0$ such that $W_\rho(H_0) = 0$. $\qquad\square$

Thus, there is no consistent test that could provide a given level of confidence under $H_0$, even if only asymptotic consistency is required under $H_1$. However, a yet weaker notion of consistency might suffice to construct asymptotically consistent clustering algorithms. Namely, we could ask for a test whose answer converges to either 0 or 1 according to whether the distributions generating the samples are independent or not. Unfortunately, it is not known whether a test consistent in this weaker sense exists or not. I conjecture that it does not. The conjecture is based not only on the result above, but also on the result of [24] that shows that there is no such test for the related problem of homogeneity testing, that is, for testing whether two given samples have the same or different distributions. This negative result holds even if the distributions are independent, binary-valued, the

difference is restricted to $P(X_0 = 0)$, and, finally, for a smaller family of processes ($B$-processes). Thus, for now what we can say is that there is no test for independence available that would be consistent under ergodic sampling. Therefore, we cannot distinguish even between the cases of 1 and 2 clusters. Thus, in the following it is assumed that the number of clusters $k$ is given.

The last problem we have to address is mutual information for processes. The analogue of mutual information for stationary processes is the mutual information rate (1). Unfortunately, 0 mutual information rate does not imply independence. This is manifest on processes with 0 entropy rate, for example those of the example in the proof of Proposition 1. What happens is that, if two processes are dependent, then indeed at least one of the $m$-order entropy rates $I_m$ is non-zero, but the limit may still be zero. Since we do not know in advance which $I_m$ to take, we will have to consider all of them, as is explained in the next subsection.

## 5.3    Clustering with the number of clusters known

The quantity introduced below, which we call sum-information, will serve as an analogue of mutual information in the i.i.d. case, allowing us to get around the problem that the mutual information rate may be 0 for a pair of dependent stationary ergodic processes. Defined in the same vein as the distributional distance (2), this new quantity is a weighted sum over all the mutual informations up to time $n$; in addition, all the individual mutual informations are computed for quantized versions of random variables in question, with decreasing cell size of quantization, keeping all the mutual information resulting from different quantizations. The latter allows us not to require the existence of densities. Weighting is needed in order to be able to obtain consistent empirical estimates of the theoretical quantity under study.

First, for a process $\mathbf{x} = (X_1, \dots, X_n, \dots)$ and for each $m, l \in \mathbb{N}$ define the $l$'th quantized version $[X_{1..m}]^l$ of $X_{1..m}$ as the index of the cell of $B^{m,l}$ to which $X_{1..m}$ belongs. Recall that each of the partitions $B^{m,l}$ has cell size $2^l$, and that $w_l := 1/l(l+1)$.

**Definition 1** (sum-information). *For stationary $\mathbf{x}_1..\mathbf{x}_N$ define the sum-information*

$$
{}^s I(\mathbf{x}_1, \dots, \mathbf{x}_N) := \sum_{m=1}^{\infty} \frac{1}{m} w_m \sum_{l=1}^{\infty} \frac{1}{l} w_l \left( \sum_{i=1}^{N} h([X_{1..m}^i]^l) \right) - h([X_{1..m}^1]^l, \dots, [X_{1..m}^N]^l) \tag{3}
$$

The next lemma follows from the fact that $\cup_{l \in \mathbb{N}} B^{m,l}$ generates $\mathcal{F}_m$ and $\cup_{m \in \mathbb{N}} \mathcal{F}_m$ generates $\mathcal{F}_\infty$.

**Lemma 1.** ${}^s I(\mathbf{x}_1, \dots, \mathbf{x}_N) = 0$ *if and only if* $\mathbf{x}_1, \dots, \mathbf{x}_N$ *are mutually independent.*

The **empirical estimates** $\hat{h}_n([X_{1..m}^i]^l)$ of entropy are defined by replacing unknown probabilities by frequencies; the estimate $\widehat{{}^s I}_n(\mathbf{x}_1, \dots, \mathbf{x}_N)$ of is obtained by replacing $h$ in (3) with $\hat{h}$.

**Remark 4** (**Computing $\widehat{{}^s I}_n$**). The expression (3) might appear to hint at a computational disaster, as it involves two infinite sums, and, in addition, the number of elements in the sum inside $h([]^l)$ grows exponentially in $l$. However, it is easy to see that, when we replace the probabilities with frequencies, all but a finite number of summands are either zero or can be collapsed (because they are constant). Moreover, the sums can be further truncated so that the total computation becomes quasilinear in $n$. This can be done exactly the same way as for distributional distance, as described in [12, Section 5].

The following lemma can be proven analogously to the corresponding statement about consistency of empirical estimates of the distributional distance, given in [22, Lemma 1].

**Lemma 2.** *Let the distribution $\rho$ of $\mathbf{x}_1, \dots, \mathbf{x}_N$ be jointly stationary ergodic. Then $\widehat{{}^s I}_n(\mathbf{x}_1, \dots, \mathbf{x}_k) \to {}^s I(\mathbf{x}_1, \dots, \mathbf{x}_N)$ $\rho$-a.s.*

This lemma alone is enough to establish the existence of a consistent clustering algorithm. To see this, first consider the following problem, which is the "independence" version of the classical statistical three-sample problem.

**The 3-sample-independence** problem. Three samples $\mathbf{x}_1, \mathbf{x}_2, \mathbf{x}_3$, are given, and it is known that *either* $(\mathbf{x}_1, \mathbf{x}_2) \perp \mathbf{x}_3$ *or* $\mathbf{x}_1 \perp (\mathbf{x}_2, \mathbf{x}_3)$ but not both. It is required to find out which one is the case.

**Proposition 2.** *There exists an algorithm for solving the 3-sample-independence problem that is asymptotically consistent under ergodic sampling.*

Indeed, it is enough to consider an algorithm that compares $\widehat{sI}_n((\mathbf{x}_1, \mathbf{x}_2), \mathbf{x}_3)$ and $\widehat{sI}_n(\mathbf{x}_1, (\mathbf{x}_2, \mathbf{x}_3))$ and answers according to whichever is smaller.

The independence clustering problem which we are after is a generalisation of the 3-sample-independence problem to $N$ samples. We can also have a consistent algorithm for the clustering problem, simply comparing all possible clusterings $U_1, \ldots, U_k$ of the $N$ samples given and selecting whichever minimizes $\widehat{sI}_n(U_1, \ldots, U_k)$. Such an algorithm is of course not practical, since the number of computations it makes must be exponential in $N$ and $k$. We will show that the number of candidate clustering can be reduced dramatically, making the problem amenable to computation.

Figure 2: CLINk: cluster given $k$ and an estimator of mutual sum-information

Consider all the clusterings obtained by applying recursively the function Split to each of the sets in each of the candidate partitions, starting with the input set $S$, until $k$ clusters are obtained. Output the clustering $U$ that minimizes $\widehat{sI}(U)$

**Function Split**(Set $S$ of samples)
Initialize: $C := \{\mathbf{x}_1\}$, $R := S \setminus C$, $\mathcal{P} := \{\}$
**while** $R \neq \varnothing$ **do**
  Initialize:$M := \{\}, d := 0$;
        xmax:= index of any $\mathbf{x}$ in $R$

  Add $(C, R)$ to $\mathcal{P}$
  **for** each $\mathbf{x} \in R$ **do**
    $r := \widehat{sI}(C, R)$
    move $\mathbf{x}$ from $R$ to $M$
    $r' := \widehat{sI}(C, R); d' := r - r'$
    **if** $d' > d$ **then**
      $d := d'$, xmax:=index of($\mathbf{x}$)
    **end if**
  **end for**
  Move $\mathbf{x}_{xmax}$ from $M$ to $C$; $R := S \setminus C$
**end while**
Return(List of candidate splits $\mathcal{P}$)
**END function**

The proposed algorithm CLINk (Algorithm 2 below) works similarly to CLIN, but with some important differences. Like before, the main procedure is to attempt to split the given set of samples into two clusters. This splitting procedure starts with a single element $\mathbf{x}_1$ and estimates its sum-information $\widehat{sI}(\mathbf{x}_1, R)$ with the rest of the elements, $R$. It then takes the elements out of $R$ one by one without replacement, each time measuring how this changes $\widehat{sI}(\mathbf{x}_1, R)$. As before, once and if we find an element that is not independent of $\mathbf{x}_1$, this change will be positive. However, unlike in the i.i.d. case, here we cannot test whether this change is 0. Yet, we can say that *if*, among the tested elements, there is one that gives a non-zero change in $sI$, then one of such elements will be the one that gives the maximal change in $\widehat{sI}$ (provided, of course, that we have enough data for the estimates $\widehat{sI}$ to be close enough to the theoretical values $sI$). Thus, we keep each split that arises from such a *maximal-change* element, resulting in $O(N^2)$ candidate splits for the case of 2 clusters. For $k$ clusters, we have to consider all the combinations of the splits, resulting in $O(N^{2k-2})$ candidate clusterings. Then select the one that minimizes $\widehat{sI}$.

**Theorem 3.** *CLINk is asymptotically consistent under ergodic sampling. This algorithm makes at most $N^{2k-2}$ calls to the estimator of mutual sum-information.*

*Proof.* The consistency of $\widehat{sI}$ (Lemma 2) implies that, for every $\varepsilon > 0$, from some $n$ on w.p. 1, all the estimates of $sI$ the algorithm uses will be within $\varepsilon$ of their $sI$ values. Since $I(U_1, \ldots, U_k) = 0$ if and only if $U_1, \ldots, U_k$ is the correct clustering (Lemma 1), it is enough to show that, assuming all the $\widehat{sI}$ estimates are close enough to the $sI$ values, the clustering that minimizes $\widehat{sI}(U_1, \ldots, U_k)$ is among those the algorithm searchers through, that is, among the clusterings obtained by applying recursively the function Split to each of the sets in each of the candidate partitions, starting with the input set $S$, until $k$ clusters are obtained.

To see the latter, on each iteration of the `while` loop, we either already have a correct candidate split in $\mathcal{P}$, that is, a split $(U_1, U_2)$ such that $sI(U_1, U_2) = 0$, or we find (executing the `for` loop) an element $\mathbf{x}'$ to add to the set $C$ such that $C \perp \mathbf{x}'$. Indeed, if at least one such element $\mathbf{x}'$ exists, then among all such elements there is one that maximizes the difference $d'$. Since the set $C$ is initialized as a singleton, a correct split is eventually found if it exists. Applying the same procedure exhaustively to each of the elements of each of the candidate splits producing all the combinations of $k$ candidate clusterings, under the assumption that all the estimates $\widehat{sI}$ are sufficiently close the corresponding values, we are guaranteed to have the one that minimizes $I(U_1, \ldots, U_k)$ among the output. $\square$

**Remark 5 (Fickle oracle).** Another way to look at the difference between the stationary and the i.i.d. cases is to consider the following "fickle" version of the oracle test of Section 3. Consider the oracle that, as before, given sets of random variables $A, B, C, D \subset \{\mathbf{x}_1, \ldots, \mathbf{x}_N\}$ answers whether $sI(A, B) > sI(C, D)$. However, the answer is only guaranteed to be correct in the case

$^sI(A, B) \neq {}^sI(C, D)$. If $^sI(A, B) = {}^sI(C, D)$ then the answer is arbitrary (and can be considered adversarial). One can see that Lemma 2 guarantees the existence of the oracle that has the requisite fickle correctness property asymptotically, that is, w.p. 1 from some $n$ on. It is also easy to see that Algorithm 2 can be rewritten in terms of calls to such an oracle.

# 6   Generalizations, future work

A general formulation of the independence clustering problem has been presented, and attempt has been made to trace out broadly the limits of what is possible and what is not possible in this formulation. In doing so, clear-cut formulations have been favoured over utmost generality, and over, on the other end of the spectrum, precise performance guarantees. Thus, many interesting questions have been left out; some of these are outlined in this section.

**Beyond time series.** For the case when the distribution of the random variables $\mathbf{x}_i$ is unknown, we have assumed that a sample $X_{1..n}^i$ is available for each $i = 1..N$. Thus, each $\mathbf{x}_i$ is represented by a time series. A time series is but one form the data may come in. Other ways include functional data, mutli-dimensional- or continuous-time processes, or graphs. Generalizations to some of these models, such as, for example, space-time stationary processes, are relatively straightforward, while others require more care. Some generalizations to infinite stationary graphs may be possible along the lines of [21]. In any case, the generalization problem is statistical (rather than algorithmic). If the number of clusters is unknown, we need to be able to replace the emulate the oracle test of section 3 with statistical tests. As explained in Section 4, it is sufficient to find a test for conditional independence, or an estimator of entropy along with guarantees on its convergence rates. If these are not available, as is the case of stationary ergodic samples, we can still have a consistent algorithm for $k$ known, as long as we have an asymptotically consistent estimator of mutual information (without rates), or, more generally, if we can emulate the fickle oracle (Remark 5).

**Beyond independence.** The problem formulation considered rests on the assumption that there exists a partition $U_1, \ldots, U_k$ of the input set $S$ such that $U_1, \ldots, U_k$ are jointly independent, that is, such that $I(U_1, \ldots, U_k) = 0$. In reality, perhaps, nothing is really independent, and so some relaxations are in order. It is easy to introduce some thresholding in the algorithms (replacing 0 in each test by some threshold $\alpha$) and derive some basic consistency guarantees for the resulting algorithms. The general problem formulation is to find a finest clustering such that $I(U_1, \ldots, U_k) > \varepsilon$, for a given $\varepsilon$ (note that, unlike in the independence case of $\varepsilon = 0$, such a clustering may not be unique). If one wants to get rid of $\varepsilon$, a tree of clusterings may be considered for all $\varepsilon \geq 0$, which is a common way to treat unknown parameters in the clustering literature (e.g.,[2]). Another generalization can be obtained by considering the problem from the graphical model point of view. The random variables $\mathbf{x}_i$ are vertices of a graph, and edges represent dependencies. In this representation, clusters are connected components of the graph. A generalization then is to clusters that are the smallest components that are connected (to each other) by at most $l$ edges, where $l$ is a parameter. Yet another generalization would be to decomposable distributions of [10].

**Performance guarantees.** Non-asymptotic results (finite-sample performance guarantees) can be obtained under additional assumptions, using the corresponding results on (conditional) independence tests and on estimators of divergence between distributions. Here it is worth noting that we are not restricted to using the mutual information $I$, but any measure of divergence can be used, for example, Rényi divergence; a variety of relevant estimators and corresponding bounds, obtained under such assumptions as Hölder continuity, can be found in [19, 11]. From any such bounds, at least some performance guarantees for CLIN can be obtained simply using the union bound over all the invocations of the tests.

**Complexity.** The algorithmic aspects of the problem have only been started upon in this work. Thus, it remains to find out what is the computational complexity of the studied problem. So far, we have presented only some upper bounds, by constructing algorithms and bounding their complexity ($kN^2$ for CLIN and $N^{2k}$ for CLIN$k$). Lower bounds (and better upper bounds) are left for future work. A subtlety worth noting is that, for the case of known distributions, the complexity may be affected by the choice of the oracle. In other words, some calculations may be "pushed" inside the oracle. In this regard, it may be better to consider the oracle for testing conditional independence, rather than a comparison of mutual informations, as explained in Remarks 1, 3. The complexity of the stationary-sampling version of the problem can be studied using the *fickle oracle* of Remark 5. The consistency of the algorithm should then be established for *every* assignment of those answers of the oracle that are arbitrary (adversarial).

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
