[Reviews · NeurIPS 2017]

Reviewer 1



The paper proposes a more general approach of independence clustering, where it considers mutual independence rather than pairwise independence. The authors provide algorithms and impossibility results given various assumptions (known/unknown distribution, known/unknown cluster size) on the problem. The first contribution (section 3, theorem 1) seems to be a naive solution to independence clustering, since it simply tests for conditional independence of a set and a variable and repeat the process. The second contribution (section 4, theorem 2) is a natural corollary of theorem 1, considering that the approximation from iid samples is a conventional approach. The third result (section 5.2, proposition 1) is more interesting in that it gives an impossibility result on stationary and ergodic process. Finally, the last result (section 5.3, theorem 3) defines sum-information which can be used as proxy to the mutual information in stationary ergodic processes. Although the convergence results are direct application of [20], the use of sum-information in independence clustering may encourage future work.

Reviewer 2



The authors study the problem of clustering random variables into the finest clusters using time-series samples. They begin with the oracle case, and propose an algorithm called CLIN, which can find the correct cluster with at most O(kn^2) conditional independence tests where k is the number of clusters and n is the number of variables. The authors then show one can make use of the existing entropy/MI estimators or conditional independence testing algorithms to implement CLIN with iid samples. For the case of stationary samples, the authors propose a similar algorithm CLINk, which requires the knowledge on the number of clusters. The results are clearly presented, and I liked reading the paper. I have few minor comments. Line 148 (h(C|R) <= h(C|R\{x}) is simply due to the property of mutual information). The first two thirds of the paper on oracle/iid samples was much easier to follows, and especially I liked the way it is written: starting with CLIN algorithm’s description and replacing the oracle estimators with sufficient estimators. For the rest of the paper, by mentioning Remark 5 upfront, one can significantly improve the readability. Finally, it will be great if the authors can add some simulation results, which corroborate their results.

Reviewer 3



ABOUT: This paper is about clustering N random variable into k mutually independent clusters. It considers the cases when k is known and unknown, as well as the cases when the distribution of the N variables is known, when it needs to be estimated from n i.i.d. samples, and when it needs to be estimated from stationary samples. Section 3 provides an optimal algorithm for known distributions which uses at most a 2kN^2 oracle calls. Subsequent sections build on this algorithm in more complicated cases. I do not know the clustering literature well enough to put this work in context. However, if the overview of the prior work is accurate, Sections 3 and 4 are a nice and important contribution. I have some reservations about Section 5. Overall, the exposition of results could be improved. COMMENTS: (1) I am not entirely convinced by the proof for Theorem 1. I believe the statement of the result is probably correct, but the proof needs to be more rigorous. It could be useful to state what exactly is meant by mutual independents for the benefit of the reader* and then to highlight in the proof how the algorithm does achieve a mutually independent clustering. For the Split function part, an inductive proof could be one way to make the proof rigorous and easier to follow. (2) It does not seem that the stationary case in Section 5 is covered adequately. It kind of reads like the authors tried to do too much and actually accomplished little. In parts, it just looks like a literature review or a math tutorial (e.g. Section 5.1). - I am not sure what the issue is with mutual information rate being zero is. One natural extension of the set up in Section 4 would be to cluster the N random variables according to their stationary distribution at time i with the n time series samples being then used to estimate this distribution as well as possible. In that case the mutual information is perfectly well defined and everything follows through. This does not seem to be what is actually being done, and instead the clustering is being performed over a whole infinite time series. - The sum information in Definition 1 seems far less interesting and fundamental than the authors try to make it out to be. Also, a side note, people have spent time thinking about what a mutual information rate for a random process should be in full generality [A] and this question maybe deserves more attention than it is given here. *I understand that some may find this too basic. But, having this definition side by side with the proof would help the reader confirm correctness. Somewhat more importantly, this issue of pairwise independence vs mutual independence is the crux of the paper. It should be stated explicitly (i.e. using math) what that means. MINOR COMMENT - Line 156: N not n [A] Te Sun Han, Information-Spectrum Method in Information Theory . Springer, 2003